# LncRNA-Encoded Micropeptides: Expression Validation, Translational Mechanisms, and Roles in Cellular Metabolism

**DOI:** 10.3390/ijms26125913

**Published:** 2025-06-19

**Authors:** Chul Woong Ho, Ji Won Lee, Chang Hoon Shin, Kyung-Won Min

**Affiliations:** 1Department of Biology, College of Natural Sciences, Gangneung-Wonju National University, Gangneung-si 25457, Republic of Korea; 2Department of Oncology Science, University of Oklahoma, Oklahoma City, OK 73104, USA

**Keywords:** micropeptides, long noncoding RNA translation, metabolic regulation

## Abstract

The discovery of functional micropeptides encoded by long noncoding RNAs (lncRNAs) has challenged the traditional view that these transcripts lack coding potential. With the advancement of high-resolution translation profiling combined with enhanced MS-based techniques, numerous lncRNAs have been found to harbor small open reading frames (sORFs) that give rise to bioactive micropeptides. These peptides participate in diverse biological processes, particularly in cellular metabolism, by modulating enzymatic activity and metabolic pathways. However, the identification and functional characterization of these micropeptides remain technically challenging due to their small size, low abundance, and the need for rigorous downstream validation studies. This review encompasses a comprehensive overview of the biogenesis of lncRNA-derived micropeptides, methodologies for detecting and validating their expression, the molecular mechanisms governing their translation, and their emerging roles in metabolic regulation. By integrating current findings and technological advancements, we highlight the potential physiological and pathological implications of these micropeptides and outline future research directions in the field.

## 1. Introduction

LncRNAs are noncoding transcripts composed of more than 200 nucleotides that play crucial roles in regulating cell proliferation and survival, often through interactions with DNA, RNA, and proteins [1,2,3]. Initially thought to be transcriptional noise in early transcriptome analyses, lncRNAs have since garnered significant attention, leading to extensive efforts to uncover their diverse functions and biological importance [4,5,6]. These lncRNAs have gained increasing recognition, partly due to findings from the ENCODE project, which revealed that a large portion of the genome is transcribed into RNA and that many of these transcripts have potential functions in regulating gene expression and other cellular processes. While protein-coding mRNAs make up only about 2% of the genome, approximately 80% of the genome shows evidence of noncoding RNAs, which are believed to be involved in regulatory activities [7,8,9]. These observations suggest that a significant portion of the genome is engaged in lncRNA expression. Recent advancements in DNA sequencing technologies have led to continuous updates in the annotation of lncRNAs, enhancing our understanding of their roles and functions [10,11].

LncRNAs play diverse roles in gene regulation, including transcriptional control, chromatin remodeling, and post-transcriptional modulation. They act as epigenetic regulators, molecular scaffolds, or microRNA sponges, influencing key biological processes such as development, cellular metabolism, and disease progression [12,13,14]. In addition to these established roles, studies have revealed that some lncRNAs can encode functional polypeptides, termed micropeptides, which contribute to cellular processes and metabolic regulation [15]. This finding challenges the long-standing view of lncRNAs as noncoding, significantly expanding our understanding of their functional diversity in the cell.

Many lncRNAs are transcribed by RNA polymerase II, undergo splicing, and are capped at the 5′ end and polyadenylated at the 3′ end, sharing similar biogenesis and structural features with mRNAs [16]. A significant number of lncRNAs also harbor small open reading frames (sORFs) [17]. However, early genome sequencing efforts faced challenges in distinguishing ORFs with true coding potential from the large number of randomly occurring, non-functional ORFs. To address this, a threshold of 100 amino acids (100 aa) was established as a practical guideline for gene annotation because statistical analyses showed that sORFs are more likely to occur randomly in the genome, making it challenging to distinguish between functional coding sequences and those arising by chance. As a result, ORFs predicted to encode peptides shorter than 100 aa were largely overlooked in genomic research for a long time [18,19]. The advent of technology such as ribosome profiling, which enables the identification of actively translated ORFs, has reshaped this view. This method has revealed that certain sORFs, even shorter than 100 aa, are associated with ribosomes. When combined with MS-based proteomics, this has led to the discovery of sORF-derived proteins, now referred to as micropeptides [20,21].

Micropeptides are commonly defined as peptides consisting of fewer than 100 amino acids; however, recent studies suggest that this criterion can be more flexibly applied. Advances in ribosome profiling, proteomics, and mass spectrometry, particularly the optimized application of improved ribosome footprint generation protocols and refined mass spectrometry workflows, have significantly enhanced the detection and annotation of small proteins [22,23]. Stable Isotope Labeling with Amino Acids in Cell Culture (SILAC) has improved quantitative accuracy, while complementary approaches such as Data-Independent Acquisition (DIA) mass spectrometry and isobaric tagging methods (iTRAQ, TMT) have expanded detection capabilities and refined the classification of micropeptides [24,25,26,27]. With these technological advancements, open reading frames (ORFs) ranging from 100 to 300 amino acids previously classified as noncoding due to the lack of translational evidence have been found to be actively translated, producing functional proteins that are now recognized as micropeptides. This challenges the traditional size-based definition and underscores the need to redefine micropeptides not solely by their length, but as small, functional proteins that were previously difficult to detect using conventional methods.

Several terms have been used in the literature to describe sORF-derived proteins, including small proteins, microproteins, small or short ORF-encoded peptides/proteins (SEPs), and micropeptides. These micropeptides may originate from lncRNAs, circular RNAs or unannotated sORFs within protein-coding genes, untranslated regions, or intergenic regions [28,29,30,31]. However, this review focuses solely on micropeptides derived from lncRNAs due to their distinct biogenesis and regulatory features, allowing for a clear and focused discussion.

Translation of micropeptides from sORFs in lncRNAs generally follows the canonical mechanisms of eukaryotic mRNA translation, including cap-dependent initiation and ribosome scanning [32,33]. However, the specific regulatory mechanisms controlling sORF translation in lncRNAs involve a non-canonical translation initiation site or interactions with specific RNA-binding proteins and noncoding RNAs [34,35]. Emerging evidence suggests that some lncRNA-derived micropeptides may be selectively translated under cellular stress conditions [36]. However, the underlying mechanisms that govern their translation and biological functions in response to cellular stress remain poorly understood. Given the complexity of translational control, further studies are needed to elucidate the regulatory mechanisms of lncRNA translation.

In this review, we summarize experimental approaches for validating the expression of lncRNA-derived micropeptides, molecular techniques employed to investigate their biological functions, and studies in understanding the regulatory mechanisms of micropeptide translation. Additionally, we highlight how these micropeptides contribute to cellular homeostasis by mediating metabolic adaptation.

## 2. Methods for Expression and Functional Validation of lncRNA-Derived Micropeptides

The discovery of sORFs within lncRNAs has raised fundamental questions regarding their potential for translation and biological significance. Ribosome profiling (Ribo-seq) analyses have revealed that many sORFs in lncRNAs are associated with ribosomes; however, determining which of these are actively translated and capable of producing micropeptides remains a significant challenge in current research [37].

### 2.1. Computational Approaches for Identifying Micropeptide-Encoding lncRNAs

The initial screening for the identification of lncRNA-derived micropeptides typically relies on computational approaches that integrate multiple layers of evidence, including sequence-based features, ribosome occupancy, evolutionary conservation, and proteomics data. Traditional gene annotation pipelines often overlooked sORFs due to their short length and presumed lack of functionality. However, advancements in bioinformatics tools have introduced novel methods for systematically identifying sORFs within lncRNAs that may encode functional micropeptides (Table 1).

One of the most used in silico approaches for predicting micropeptide-encoding lncRNAs is a sequence-based coding potential assessment. Tools such as Coding Potential Calculator 2 (CPC2) and Coding Potential Assessment Tool (CPAT) evaluate nucleotide composition, codon usage bias, and ORF length to differentiate coding from noncoding transcripts [38,39]. These tools serve as an initial filter for identifying candidate lncRNAs that may encode small peptides. However, since many sORFs fall below the conventional size thresholds used in gene annotation, additional computational strategies are required to refine these predictions.

Ribo-seq data has proven instrumental in identifying actively translated sORFs. Computational tools such as ORF-RATER and RiboTISH leverage Ribo-seq coverage patterns to assess the likelihood of translation initiation and ribosome engagement [40,41]. These approaches have demonstrated that many lncRNAs harbor translatable sORFs, challenging the conventional view that lncRNAs function solely as regulatory RNAs. However, not all translated sORFs give rise to stable and functional micropeptides, making it necessary to integrate additional evidence, such as evolutionary conservation and proteomics validation.

Evolutionary conservation analysis provides further insights into the functional relevance of sORFs. Highly conserved sORFs are more likely to encode biologically active micropeptides, suggesting they are not merely byproducts of random transcriptional noise or the result of randomly occurring sequences in the genome. Tools such as PhyloCSF and RNAcode analyze codon substitution patterns across species to assess the coding potential of a given sORF [42,43,44]. While evolutionary conservation is a strong indicator of functionality, it is important to note that species-specific micropeptides may still exist despite a lack of cross-species conservation. For example, *LINC00467* produces functional micropeptides, yet it is classified as a noncoding RNA based on PhyloCSF analysis, indicating that conservation analysis alone may not fully capture the functional potential of sORFs, although it can complement other approaches [45].

**Table 1 ijms-26-05913-t001:** In silico methods for identifying micropeptide-encoding lncRNAs.

Tool	Function	Website	Ref.
CPC2	Predicts the coding potential of input RNA sequences using a machine learning model.	https://cpc2.gao-lab.org/ (accessed on 17 June 2025)	[38]
CPAT	Evaluates coding potential based on ORF length, Fickett score, and codon usage bias.	https://code.google.com/archive/p/cpat/ (accessed on 17 June 2025)	[39]
ORF-RATER	Analyzes ribosome occupancy patterns from Ribo-seq data to assess the translation potential of ORFs.	https://github.com/alexfields/ORF-RATER (accessed on 17 June 2025)	[40]
RiboTISH	A Ribo-seq-based tool for identifying translation initiation sites (TISs), allowing precise localization of start codons.	https://github.com/zhpn1024/ribotish (accessed on 17 June 2025)	[41]
PhyloCSF	Predicts whether an sORF encodes a conserved protein by analyzing codon substitution patterns across multiple species.	https://github.com/mlin/PhyloCSF (accessed on 17 June 2025)	[42,44]
RNAcode	Identifies evolutionarily conserved coding regions by analyzing conserved codon patterns from multiple sequence alignments.	https://github.com/ViennaRNA/RNAcode (accessed on 17 June 2025)	[43]
PeptideAtlas	A mass spectrometry-based protein/peptide detection database that includes experimentally verified micropeptides.	https://peptideatlas.org/ (accessed on 17 June 2025)	[46]
OpenProt	A comprehensive database providing information on non-canonical ORFs (e.g., alternative ORFs, sORFs) and their potential protein products, extending beyond standard annotations.	https://www.openprot.org/ (accessed on 17 June 2025)	[47]

Incorporating proteomics data into computational predictions further strengthens the identification of functional micropeptides. Databases such as PeptideAtlas and OpenProt store experimentally detected micropeptides, allowing researchers to cross-reference computational predictions with mass spectrometry-based evidence. Integrative proteogenomic approaches, which combine RNA-seq, Ribo-seq, and mass spectrometry, offer a powerful framework for discovering and validating novel micropeptides derived from lncRNAs [46,47].

Although in silico methods have greatly expanded our ability to predict micropeptide-encoding lncRNAs, experimental validation remains essential for confirming their expression and functional relevance.

### 2.2. In Vitro Validation of Micropeptide Expression

In vitro validation of micropeptide expression focuses on evaluating whether a predicted sORF is actively translated using ribosome association assays, peptide detection techniques, translation reporter assays, and immunodetection methods (Figure 1).

Ribo-seq and polysome profiling are two commonly used methods for verifying micropeptide expression. Ribo-seq analyzes ribosome-protected fragments (RPFs), providing direct evidence of translation initiation and elongation. Polysome profiling, on the other hand, fractionates polysome-associated RNA to identify actively translated transcripts. While Ribo-seq offers a detailed view of translation activity, polysome profiling focuses on translational efficiency by detecting transcripts associated with multiple ribosomes. These methods can provide complementary insights, as Ribo-seq maps ribosome occupancy, while polysome profiling highlights translation efficiency. Polysome profiling followed by qPCR is particularly useful for evaluating individual lncRNA translation, making these approaches valuable for confirming the translation of micropeptides (Figure 1A,B) [48,49,50].

Mass spectrometry (MS)-based proteomics is also widely employed to directly confirm the presence of micropeptides (Figure 1C). Techniques like shotgun proteomics and targeted proteomics (e.g., Selected Reaction Monitoring, SRM, and Parallel Reaction Monitoring, PRM) enable the highly sensitive detection and quantification of small peptides. These methods provide detailed information on peptide identification, sequences, and abundance [51,52,53].

An alternative approach for confirming micropeptide expression involves the development of antibodies against micropeptides. However, designing such antibodies can be particularly challenging due to the short length and low expression levels of micropeptides. As a result, reporter gene-based approaches including epitope tagging (e.g., FLAG, HA), luciferase reporters, and fluorescent protein fusions (e.g., GFP, RFP) are frequently employed to validate the translation of predicted micropeptides (Figure 1D). Luciferase reporters are particularly useful for quantifying translation efficiency through luminescence signals, while fluorescent proteins enable real-time visualization of expression levels and subcellular localization in live cells. Epitope tags, due to their small size, are advantageous for preserving the structural and functional integrity of micropeptides during detection via techniques such as Western blotting (WB), immunoprecipitation (IP), and immunofluorescence (IF) assays. Additionally, CRISPR-Cas9-mediated knock-in of endogenous tags enables the tracking of micropeptide expression under physiological conditions, eliminating the need for exogenous expression vectors and ensuring more accurate representation of natural expression levels (Figure 1E) [54,55].

### 2.3. Molecular Approaches for Understanding Micropeptide Functions

The presence of lncRNA-derived micropeptides has been supported by computational predictions and validated through in vitro experiments. However, a comprehensive understanding of their biological functions requires further molecular-level investigations.

Loss-of-function and gain-of-function experiments provide essential insights into the roles of micropeptides. CRISPR-Cas9-mediated knockout and RNA interference (RNAi)-based knockdown approaches can be used to evaluate the impact of micropeptide loss on cellular function [55,56,57,58]. However, these strategies may inadvertently affect the function of the host lncRNA, resulting in phenotypic changes that are not solely attributable to the micropeptides. Therefore, more precise methods such as single-nucleotide substitutions that specifically remove the start codon of the micropeptides are required [59]. This approach enables selective inhibition of micropeptide translation while preserving the transcriptional function of the lncRNA. Conversely, overexpression experiments using lentiviral vectors or inducible plasmids can assess how increased micropeptide expression influences cellular proliferation, apoptosis, or differentiation. These methods help determine whether a specific micropeptide is essential for maintaining cellular homeostasis (Figure 2A). For instance, HOXB-AS3, a micropeptide encoded by the *HOXB-AS3* lncRNA, has been validated through functional studies utilizing CRISPR-Cas9 knockout and overexpression assays, which have confirmed its role in the regulation of tumor growth [60]. Another notable example is SPAR (Small Regulatory Peptide of Amino Acid Response), derived from *LINC00961*. RNAi knockdown experiments have demonstrated that the depletion of SPAR results in disrupted muscle homeostasis, highlighting its critical regulatory function [61].

Due to their small size, micropeptides typically have limited functionality on their own and mainly exert their effects by interacting with other proteins, thereby participating in various cellular processes. As such, analyses of protein interactions are crucial for understanding their functions. Several experimental techniques are used to uncover these interactions. Co-immunoprecipitation (Co-IP) followed by mass spectrometry or Western blot analysis is commonly employed to identify proteins that bind to micropeptides (Figure 2B) [62]. However, this method mainly detects stable interactions and may miss transient or weak associations. To overcome these limitations, proximity labeling techniques have been developed [63]. These methods enable the real-time tagging of proteins located near the micropeptides within living cells, facilitating interaction network analysis. Typically, the micropeptides are fused to proximity labeling enzymes, such as a biotin ligase (BioID or TurboID) or a peroxidase (APEX2) (Figure 2C) [64,65,66]. Notably, TurboID offers faster reaction kinetics and higher efficiency than BioID or APEX2, effectively labeling nearby proteins in as little as 10 min at room temperature. Upon activation, these enzymes catalyze the biotinylation of nearby proteins, enabling their enrichment through streptavidin-based affinity purification followed by identification via mass spectrometry. For example, Chu et al. utilized the APEX2 proximity labeling technique to demonstrate that the 123-amino acid micropeptide encoded by *C11orf98* sORF is in close proximity to nucleophosmin (NPM1) and nucleolin (NCL) [67].

While functional studies have begun to reveal the physiological significance of lncRNA-derived micropeptides, the mechanisms by which these peptides are selectively translated remain poorly understood. Given that many of these micropeptides are produced under stress or specific cellular contexts, it is essential to investigate the regulatory processes that control their translation. In the following section, we discuss emerging insights into the translational mechanisms that enable the selective production of micropeptides, particularly under conditions where conventional cap-dependent translation is inhibited.

## 3. Mechanisms Regulating the Translation of lncRNA-Derived Micropeptides

While some lncRNAs are translated through the conventional cap-dependent mechanism, growing evidence suggests that lncRNA-derived micropeptides often rely on cap-independent pathways (Figure 3A). Notably, the lack of a positive correlation between lncRNA expression levels and the translation of their encoded micropeptides supports the existence of alternative translation mechanisms [37]. These pathways may be particularly active under cellular conditions where global cap-dependent translation is suppressed.

One well-characterized cap-independent translation mechanism is internal ribosome entry site (IRES)-mediated translation, which facilitates ribosome recruitment and translation initiation at internal regions of the RNA without relying on eIF4E-mediated 5′-cap recognition [35,68]. This mechanism is often activated under stress conditions, such as apoptosis or DNA damage, allowing selective translation of specific mRNAs involved in stress adaptation and survival [68,69,70]. This suggests that lncRNAs harboring sequence motifs that function as IRES elements may also undergo translation under such conditions. For instance, Yu et al. demonstrated that the DDUP micropeptides (186 aa), encoded by lncRNA *CTBP1-DT*, are translated through an IRES-mediated mechanism in response to DNA damage [71]. Under normal conditions, two upstream open reading frames (uORFs) within *CTBP1-DT* suppress DDUP translation by interfering with ribosome scanning. However, in response to DNA damage, ribosomes engage with the IRES region and bypass these uORFs, enabling cap-independent translation of DDUP. Once synthesized, DDUP enhances homologous recombination repair (HRR) and post-replication repair (PRR) by stabilizing the RAD18/RAD51C and RAD18/PCNA complexes, respectively. As a result, DDUP increases the sensitivity of ovarian cancer cells to DNA-damaging chemotherapies (Figure 3B).

Another cap-independent translation mechanism involves N6-methyladenosine (m6A) methylation, particularly at a conserved site in 5′UTR [34]. m6A modifications can promote translation by recruiting reader proteins such as YTHDF1 and YTHDF3, which facilitate ribosome assembly independent of the 5′ cap by recruiting the translation initiation factor eIF3 [34]. In non-small-cell lung cancer (NSCLC), Pei et al. demonstrated that m6A modification at the 1313 adenine regulates translation of the micropeptide ATMLP (90 aa) encoded by lncRNA *AFAP1-AS1* [72]. Enhanced translation of ATMLP contributes to tumor progression by interfering with mitophagy and promoting oncogenic functions. Similarly, Wu et al. demonstrated that m6A is essential for the translation of YY1BM, a micropeptide encoded by lncRNA *LINC00278*, in esophageal squamous-cell carcinoma (ESCC) [73]. This process is regulated by m6A writer proteins METTL3, METTL14, and WTAP, which enhance translation by recruiting YTHDF1. Notably, cigarette smoking increases the expression of ALKBH5, an m6A demethylase, resulting in reduced m6A modification and decreased YY1BM translation, which promotes ESCC progression (Figure 3C).

Furthermore, our previous study demonstrated that MST1 phosphorylates eIF4E at threonine 55 (T55), reducing its affinity for the 5′ cap and thereby suppressing cap-dependent translation of a subset of mRNAs [36]. Interestingly, eIF4E phosphorylation enhances translation of lncRNA *LINC00689*, which encodes the micropeptide STORM (50 aa). MST1 activation by TNF-α induces eIF4E phosphorylation and increases the association of *LINC00689* with polyribosomes, promoting STORM production under conditions in which the cap-binding affinity of eIF4E is reduced (Figure 3D). Functionally, STORM exhibits molecular mimicry of the signal recognition particle protein SRP19, competing with *7SL* RNA binding and thereby disrupting SRP complex formation and modulating protein targeting to the endoplasmic reticulum. These findings highlight the emerging regulatory roles of micropeptides, underscoring the need for further investigation to uncover the precise molecular mechanisms governing their translation (Figure 3).

## 4. Regulatory Roles of lncRNA-Encoded Micropeptides in Cellular Metabolism

Metabolic adaptation and reprogramming occur not only in cancer cells but also in diverse cell types and pathological conditions, where they play a key role in cellular function and homeostasis [74,75,76,77]. However, understanding the fine-tuning of metabolic processes has not been fully characterized by the activity of protein-coding mRNAs alone, suggesting that there could be another layer of regulation involving additional factors. Recent studies suggest that lncRNA-encoded micropeptides serve as key regulators of metabolic reprogramming via diverse mechanisms. Herein, we summarize the role of these micropeptides in different metabolic processes (Figure 4 and Table 2).

Glucose uptake is the initial and rate-limiting step of glucose metabolism by ensuring a sufficient glucose supply for glycolysis and other metabolic pathways. Glucose transporter 1 (GLUT1) and glucose transporter 3 (GLUT3) not only facilitate glucose uptake but also play a crucial role in supporting early embryogenesis [78]. Fu et al. demonstrated that Nodal signaling, a key regulator of gastrulation and mesendoderm differentiation, also regulates glucose metabolism via NEMEP, a micropeptide (63 aa) encoded by lncRNA *Gm11549* in mouse embryonic stem cells (mESCs) [79]. NEMEP enhances glucose uptake by interacting with GLUT1 and GLUT3, and its depletion significantly reduces glucose influx, impairing mesendoderm differentiation.

Once glucose enters the cell, it undergoes glycolysis, a tightly regulated process that converts glucose into pyruvate while generating ATP. Pyruvate kinase M (PKM) is a key regulator of glycolytic flux; it catalyzes the conversion of phosphoenolpyruvate to pyruvate, thereby controlling the final step of glycolysis. Alternative splicing of PKM pre-mRNA generates two isoforms, PKM1 and PKM2, which dictate distinct metabolic fates. PKM1 is predominantly expressed in differentiated tissues and promotes oxidative phosphorylation, whereas PKM2 is highly expressed in proliferating cells and cancer cells, favoring aerobic glycolysis to support biosynthetic processes [80,81,82]. Recent studies revealed that this splicing process is regulated by lncRNA-encoded micropeptides, which modulate PKM isoform expression and influence cellular metabolism. For instance, the HOXB-AS3 micropeptide (53 aa), encoded by lncRNA *HOXB-AS3*, functions as a tumor suppressor in colorectal cancer (CRC) [60]. It suppresses PKM2 translation, thereby shifting cellular metabolism from glycolysis to oxidative phosphorylation. Mechanistically, HOXB-AS3 interacts with heterogeneous nuclear ribonucleoprotein A1 (hnRNP A1), a well-established splicing regulator, to modulate the expression of PKM isoforms. By preventing hnRNP A1-mediated splicing of *PKM* pre-mRNA, HOXB-AS3 suppresses PKM2 production while promoting PKM1 expression. This metabolic shift reduces glucose consumption and ATP production through glycolysis, ultimately suppressing cancer cell proliferation and tumor growth.

Similarly, micropeptide GMRSP (131 aa), which is encoded by lncRNA *H19*, has been identified as a key modulator of metabolic reprogramming in vascular smooth muscle cells (VSMCs). GMRSP functions by interacting with heterogenous nuclear ribonucleoprotein A2B1 (hnRNP A2B1), which is another splicing regulator involved in PKM isoform selection. GMRSP regulates the expression of PKM2 by interacting with hnRNP A2B1, thereby inhibiting the binding of hnRNP A2B1 to *PKM* pre-mRNA. This interaction leads to decreased PKM2 expression and increased PKM1 expression. Consequently, this alteration impacts glucose metabolism and may redirect metabolic flux toward oxidative phosphorylation [83].

In addition to glycolysis and lactate metabolism, cells utilize the pentose phosphate pathway (PPP) to generate NADPH and ribose-5-phosphate (R5P), which are essential for redox homeostasis and nucleotide biosynthesis [84]. The micropeptide pep-AP (37 aa) modulates chemotherapy sensitivity in colorectal cancer by regulating the PPP [85]. Mechanistically, pep-AP encoded by lncRNA *lnc-AP* interacts with transaldolase 1 (TALDO1), a key enzyme in the PPP, reducing NADPH/NADH+ and glutathione (GSH) levels. This disruption promotes the accumulation of reactive oxygen species (ROS) and ultimately induces apoptosis, thereby enhancing the sensitivity of CRC cells to treatments such as Oxaliplatin chemotherapy.

Mitochondrial oxidative phosphorylation (OXPHOS) is a key pathway for ATP production, and lncRNA-encoded micropeptides play an important role in its regulation. Ge et al. identified ATP synthase-associated peptide (ASAP), a micropeptide (94 aa) encoded by lncRNA *LINC00467*, as a critical regulator of OXPHOS in CRC [86]. ASAP localizes to the inner mitochondrial membrane (IMM) and directly interacts with ATP synthase subunits a (ATP5A) and γ (ATP5C), enhancing their interaction and promoting ATP synthase activity. This results in increased mitochondrial ATP production and a higher oxygen consumption rate (OCR), ultimately supporting CRC cell proliferation and tumor growth.

*LINC00116*-derived micropeptide Mitoregulin (MTLN) (56 aa) was initially reported to localize to the mitochondrial inner membrane (IMM) and regulate fatty acid oxidation and oxidative phosphorylation [87,88]. However, Zhang et al. provided experimental evidence that MTLN is localized to the mitochondrial outer membrane (OMM) instead of the IMM using multiple orthogonal methods [89]. MTLN interacts with carnitine palmitoyltransferase 1B (CPT1B) and cytochrome b5 type B (CYB5B), two key regulator enzymes for mitochondrial fatty acid metabolism. Knockdown of MTLN leads to the accumulation of very-long-chain fatty acids (VLCFAs), particularly docosahexaenoic acid (DHA), which impairs fatty acid oxidation and consequently disrupts lipid metabolic flux. These findings establish MTLN as a key regulator of fatty acid metabolism at the outer membrane of mitochondria.

Similarly, pep-AKR1C2, a micropeptide (163 aa) encoded by the exosomal lncRNA *lncAKR1C2*, has been identified as a key regulator of fatty acid oxidation (FAO) and lymphatic metastasis in gastric cancer (GC) [90]. Following exosome-mediated transfer from GC cells to recipient human lymphatic endothelial cells (HLECs), *lncAKR1C2* is translated into pep-AKR1C2, which subsequently promotes FAO by upregulating CPT1A. Mechanistically, pep-AKR1C2 enhances the activity of Yes-associated protein (YAP) by reducing its phosphorylation, promoting its nuclear translocation and subsequent activation of *CPT1A* transcription. This shift in lipid metabolism increases ATP production and lymphangiogenesis, thereby fueling lymph node metastasis (LNM) in GC.

Ferroptosis is a lipid peroxidation-driven form of cell death characterized by mitochondrial shrinkage, increased membrane density, and the accumulation of iron-dependent reactive oxygen species (ROS) [91]. Tong et al. identified HCP5-132aa, a micropeptide encoded by the lncRNA *HCP5*, as a regulator of ferroptosis in triple-negative breast cancer (TNBC) [92]. HCP5-132aa may prevent ferroptosis by regulating glutathione peroxidase 4 (GPX4), a critical enzyme that prevents lipid peroxidation [93]. HCP5-132aa is highly expressed and promotes malignant phenotypes of TNBC cell lines. To verify its regulatory mechanism, they confirmed that knockdown of HCP5-132aa reduces GPX4 expression, increases ROS levels, and promotes lipid peroxidation, which may enhance ferroptotic cell death in TNBC cells. Additionally, xenograft models have demonstrated that loss of HCP5-132aa inhibits tumor growth, highlighting its potential as a prognostic biomarker. Taken together, lncRNA-encoded micropeptides play a vital role in regulating cellular metabolism by modulating key pathways such as glucose uptake, glycolysis, oxidative phosphorylation, fatty acid oxidation, and ferroptosis. These findings highlight the emerging complexity of metabolic reprogramming, where micropeptides offer an additional layer of regulation beyond traditional protein-coding genes.

**Table 2 ijms-26-05913-t002:** Overview of lncRNA-encoded micropeptides and their roles in cellular metabolism.

No.	LncRNA	Micropeptide	Size (aa)	Metabolism	Interacting Protein	Cancer/Cell Type	Function and Mechanism	Ref.
1	*Gm11549*	NEMEP	63 aa	Glucose metabolism	GLUT1/3	Embryonic stem cells	Interacts with GLUT1 and GLUT3, enhancing glucose uptake.	[79]
2	*HOXB-AS3*	HOXB-AS3	53 aa	Glucose metabolism	hnRNP A1	Colorectal cancer	Suppresses PKM2 isoform by interacting with hnRNP A1, leading to a metabolic shift from glycolysis to oxidative phosphorylation.	[60]
3	*H19*	GMRSP	131 aa	Glucose metabolism	hnRNP A2B1	Vascular smooth muscle cell	Interacts with hnRNP A2B1 to suppress PKM2 isoform, modulating metabolic flux toward oxidative phosphorylation.	[83]
4	*Lnc-AP*	pep-AP	37 aa	Pentose phosphate pathway	TALDO1	Colorectal cancer	Interacts with TALDO1, reducing NADPH and glutathione levels, inducing apoptosis, and enhancing chemotherapy sensitivity.	[85]
5	*LINC00116*	MTLN	56 aa	Fatty acid metabolism	CPT1B, CYB5B	Human Embryo Kidney cells	Interacts with CPT1B and CYB5B at the mitochondrial outer membrane, regulating fatty acid oxidation and lipid metabolic flux.	[89]
6	*LncAKR1C2*	pep-AKR1C2	163 aa	Fatty acid metabolism	CPT1A	Gastric cancer	Activates YAP signaling, which increases CPT1A expression and promotes fatty acid oxidation, leading to higher ATP production and facilitating lymphatic metastasis.	[90]
7	*LINC00467*	ASAP	94 aa	Mitochondrial activity	ATP5A and ATP5C	Colorectal cancer	Enhances interaction between ATP5A and ATP5C, promoting ATP synthase activity and mitochondrial oxygen consumption.	[86]
8	*HCP5*	HCP5-132aa	132 aa	Glutathione metabolism (ferroptosis)	GPX4	Triple-negative breast cancer	Regulates ferroptosis by modulating GPX4 expression, which increases ROS levels and lipid peroxidation.	[92]

## 5. Conclusions

Recent advances in transcriptomics, ribosome profiling, and proteogenomics have revealed that a subset of lncRNAs contain translatable small open reading frames (sORFs), giving rise to functional micropeptides. These findings challenge the classical notion that lncRNAs are purely regulatory, noncoding molecules. Growing evidence suggests that lncRNA-derived micropeptides are crucial regulators of cellular metabolism. In the context of metabolic regulation, micropeptides have been shown to act as critical modulators of major pathways, including glucose uptake, glycolysis, oxidative phosphorylation, fatty acid oxidation, and ferroptosis.

Despite their small size, micropeptides can exert substantial biological functions, often through interactions with protein complexes or participating in signaling pathways. Their small size and relatively simple structure may allow them to respond more swiftly than larger proteins to various cellular stresses, such as those arising from disease or metabolic challenges. On the other hand, their minimalistic structure may suggest that they perform highly specific roles in pathological contexts, rather than exhibiting broad multifunctionality.

Consequently, micropeptides hold significant promise as potential therapeutic targets. Further exploration of their functional roles and translational mechanisms could open new avenues for developing innovative strategies to combat disease including cancer.

## 6. Perspectives

Research on lncRNA-encoded micropeptides is still at an early stage, and much about their biological properties remains largely uncharacterized. To date, most studies have focused on confirming the expression and biological functions of micropeptides. In contrast, the molecular mechanisms governing how their translation is selectively regulated under specific stress conditions remain poorly understood. Future research should place greater emphasis on elucidating the regulatory mechanisms underlying this selective translation.

Efforts to predict the existence of micropeptides often rely on computational tools that incorporate sequence features, ribosome occupancy, evolutionary conservation, and proteomics data. Given that lncRNA biogenesis shares similarities with mRNA, such as 5′ capping and 3′ polyadenylation, these structural characteristics may also serve as supportive indicators of translational potential for lncRNA-encoded micropeptides. Although the biogenesis and maturation processes of mRNA and lncRNA appear similar, there are notable differences. For example, lncRNA-encoded micropeptides generally possess multiple short exons compared to mRNAs, which could influence splicing efficiency, as lncRNAs often harbor weaker and less conserved splice sites [94]. Additionally, improperly matured RNA is typically retained in the nucleus [95], and the presence of multiple open reading frames can activate the nonsense-mediated RNA decay (NMD) pathway [96], potentially hindering lncRNA translation. Despite these challenges, some lncRNA-encoded micropeptides evade NMD and are successfully translated, emphasizing the need for further investigation into their regulatory mechanisms.

Advances in emerging technologies now allow for more precise and comprehensive interrogation of protein expression and function. Applying these techniques to micropeptide research is expected to greatly enhance our understanding of their biological significance and functional context. Recent advances in biological nanopore technology have enabled the discrimination of individual amino acids and short peptides based on subtle changes in electrical signals, suggesting its potential applicability for the direct detection and post-translational modification (PTM) analysis of small proteins, which typically range from 10 to 100 amino acids in length. In particular, nanopore sensing such as EXP2 and FraC has gained attention as a promising platform due to its structural versatility and high electrical sensitivity [97,98]. For example, recent developments in nanopore-based peptide sequencing are revolutionizing the identification of micropeptides. Iesu et al. demonstrated single-molecule resolution capable of distinguishing cis/trans proline isomers, underscoring the potential of nanopores for high-precision peptide analysis [99]. These nanopores can detect subtle changes in ionic current as individual peptides translocate through the pore, providing a label-free, single-molecule method for analyzing small proteins such as micropeptides.

To precisely define the physiological and functional relevance of micropeptides, it is essential to determine their spatial expression patterns within specific tissues or cell types. Spatial transcriptomics has emerged as a powerful tool for visualizing the expression patterns of lncRNAs in situ, enabling the identification of the spatially and conditionally restricted expression of micropeptide-encoding lncRNAs under various contexts such as tumor microenvironments or metabolic stress [100]. Given that many lncRNAs exhibit tissue-specific or stress-responsive expression patterns, such spatial information provides key insights into the functional context of their encoded micropeptides. However, RNA abundance does not necessarily reflect protein expression or function. Therefore, spatial proteomics approaches are also essential. Recent advances, such as Deep Visual Proteomics (DVP), which integrates high-resolution imaging, AI-based cell segmentation, and ultrasensitive mass spectrometry, allow for direct, unbiased analysis of the proteomic landscape at a single-cell resolution within defined spatial contexts [101]. The integration of spatial transcriptomic and proteomic data will be instrumental in elucidating the precise localization, functional roles, and context-dependent regulation of lncRNA-derived micropeptides under in situ conditions.

## Figures and Tables

**Figure 1 ijms-26-05913-f001:**
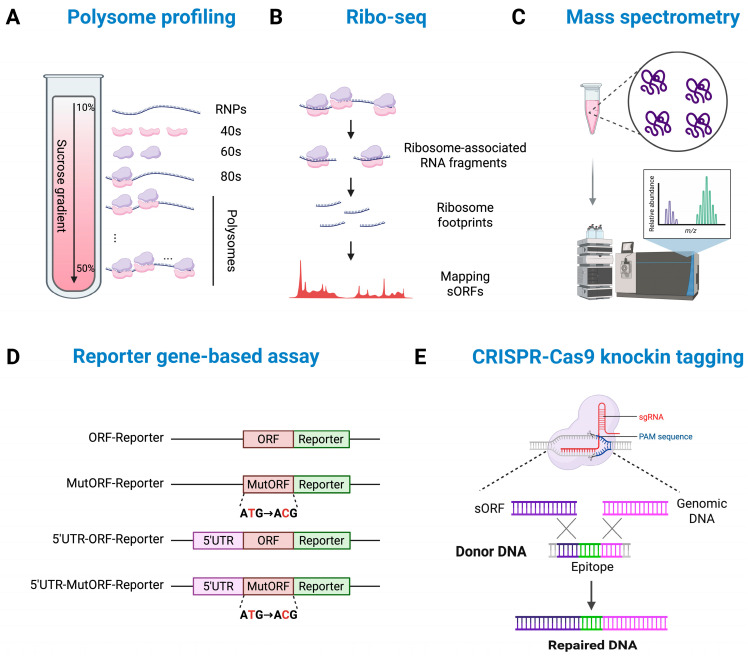
In vitro validation of lncRNA-derived micropeptide expression: This diagram summarizes representative methods used to identify and validate micropeptides encoded by lncRNAs. (**A**) Polysome profiling for assessing the association of lncRNA with polysome. (**B**) Ribosome profiling (Ribo-seq) to identify actively translated sORFs. (**C**) Mass spectrometry-based detection and validation of micropeptides. (**D**) Reporter gene-based assay to verify translation of sORFs. (**E**) CRISPR-Cas9-mediated knock-in for endogenous tags, enabling detection of micropeptides via Western blot (WB) and immunofluorescence (IF).

**Figure 2 ijms-26-05913-f002:**
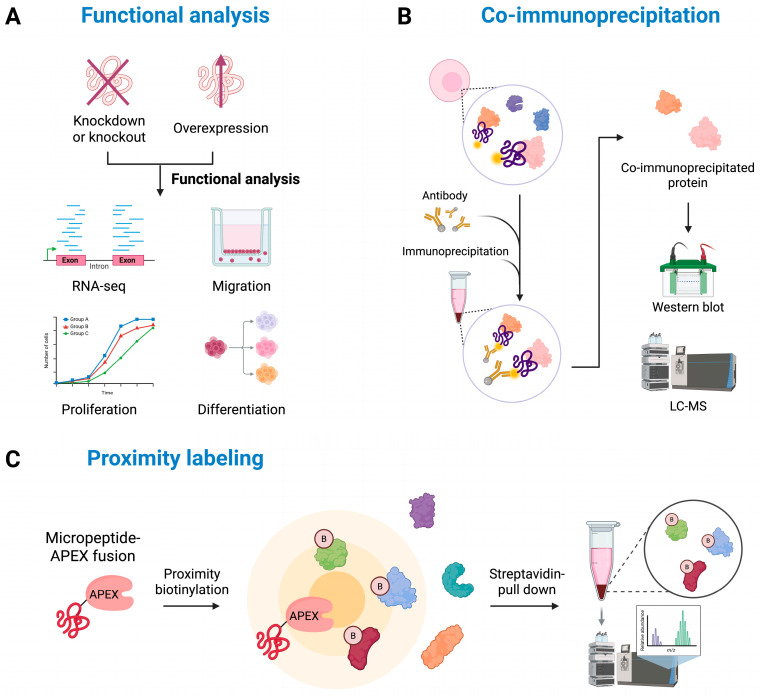
In vitro functional analysis of lncRNA-derived micropeptides: This diagram summarizes representative methods used to investigate the functions and interactions of micropeptides encoded by lncRNAs. (**A**) Functional characterization through loss- and gain-of-function experiments using RNA interference (KD), CRISPR-Cas9-mediated knockout or start codon mutation (KO), and overexpression (OE) of sORFs, followed by evaluation of transcriptomic alterations and cellular phenotype (e.g., proliferation, differentiation). (**B**) Co-immunoprecipitation (Co-IP) of epitope-tagged micropeptides followed by mass spectrometry and Western blot analysis to identify interacting proteins. (**C**) Proximity labeling using APEX-tagged micropeptide reporters to biotinylate nearby proteins, enabling purification and identification of micropeptide-interacting proteins by mass spectrometry.

**Figure 3 ijms-26-05913-f003:**
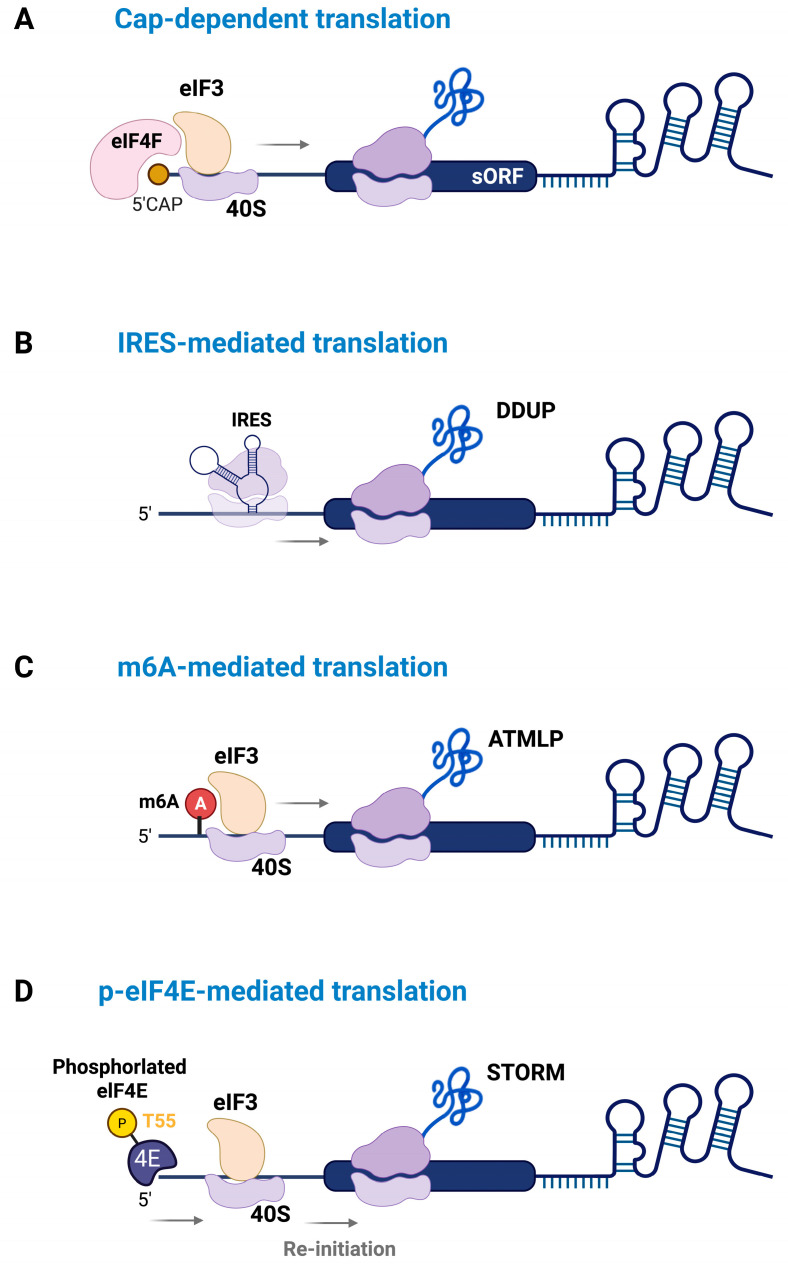
Translation initiation mechanisms of lncRNA-derived micropeptides: The diagram illustrates how micropeptides are translated via (**A**) cap-dependent translation involving the canonical eIF4E complex, (**B**) internal ribosome entry site (IRES)-mediated translation, (**C**) m6A-mediated cap-independent translation, and (**D**) phosphorylation of elF4E, which selectively enhances the translation of lncRNAs.

**Figure 4 ijms-26-05913-f004:**
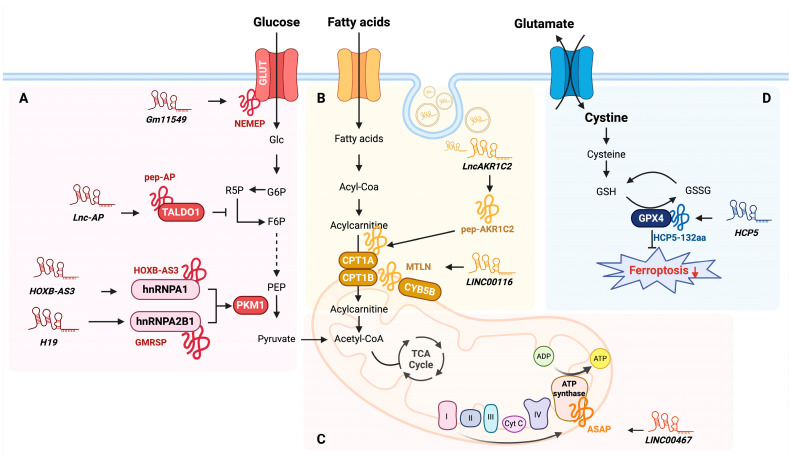
Regulatory roles of lncRNA-encoded micropeptides in cellular metabolism: Schematic illustration showing the involvement of lncRNA-encoded micropeptides in metabolic pathways. The diagram details how these micropeptides regulate (**A**) glucose metabolism, (**B**) fatty acid metabolism, (**C**) mitochondrial oxidative phosphorylation, and (**D**) ferroptosis via interactions with key metabolic regulators. Solid arrows indicate direct metabolic flows or molecular interactions, while dotted arrows represent multi-step processes. Red arrow indicates suppression of the corresponding biological process, and blunt-ended arrows indicate inhibition. Roman numerals I–IV in panel (C) refer to mitochondrial respiratory chain complexes (Complex I–IV).

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
