# Peer review of "LncRNA-Encoded Micropeptides: Expression Validation, Translational Mechanisms, and Roles in Cellular Metabolism"

_ijms, 2025, doi:10.3390/ijms26125913_

Round 1
Reviewer 1 Report
Comments and Suggestions for Authors
The manuscript titled “Lnc-RNA Encoded Micropeptides: Expression, Validation, Translational Mechanisms, and Roles in Cellular Metabolism” by Chul Woong Ho and colleagues is a great review that discusses the recent understanding in the field of expression of micropeptides encoded by lncRNAs. The authors have highlighted the methodologies used to identify such micropeptides, the regulatory mechanisms involved in their production, and their involvement in the metabolic health of cells. The filed of micropeptides is expected to boom within the next decade or so, so the review is a very timely study on the state of the field.
The manuscript is well written; it contains logical structure and makes for an enjoyable reading for experts in the field. The writing is appropriate for non-experts with a reasonable scientific background as well. The paragraph sectioning and the flow makes sense. As a matter of fact, this reviewer was left wanting to know more after finishing reading the manuscript, so all my recommendations have to do with potential additions to the paper:
- A definition of micropeptides based on size. Some of the discussion involves peptides that are 132 residues long, are those considered micropeptides?
- Some background on the history of gene and protein characterization would be warranted early on in the paper. It is my understanding that at the time of gene cloning (at its peak in the 80s and 90s), any ORFs predicted to generate short peptides (<100 residues maybe?) were disregarded as not functional, despite the fact that short peptides, such as the atrial natriuretic peptide, were already known to have functions and were considered outliers at the time. The same goes for Xist and lncRNAs. This sort of perspective would make the paper more informative in terms of understanding what the limitations and barriers have been in the field of identifying and characterizing micropeptides in the past.
- The authors do not discuss the potential of circRNAs to generate peptides, which has been a proposed functional mechanism for this group of lncRNAs.
- There are a couple of published papers that were likely not yet out when the authors were preparing their manuscript, but they describe recent advances in nanopore-based sequencing of peptides. One was published in Chemical Science by a French group, and one was a review by a Dutch group published in Nature Biotechnology. Discussion of these papers in the “Perspectives” section would make the manuscript more current. Full disclosure: I am not affiliated with either of these groups.
Author Response
Reviewer 1:
The manuscript titled “Lnc-RNA Encoded Micropeptides: Expression, Validation, Translational Mechanisms, and Roles in Cellular Metabolism” by Chul Woong Ho and colleagues is a great review that discusses the recent understanding in the field of expression of micropeptides encoded by lncRNAs. The authors have highlighted the methodologies used to identify such micropeptides, the regulatory mechanisms involved in their production, and their involvement in the metabolic health of cells. The filed of micropeptides is expected to boom within the next decade or so, so the review is a very timely study on the state of the field.
The manuscript is well written; it contains logical structure and makes for an enjoyable reading for experts in the field. The writing is appropriate for non-experts with a reasonable scientific background as well. The paragraph sectioning and the flow makes sense. As a matter of fact, this reviewer was left wanting to know more after finishing reading the manuscript, so all my recommendations have to do with potential additions to the paper:
[AU] We appreciate the reviewer’s evaluation. We have addressed each comment accordingly, as detailed below. The corresponding revisions have also been highlighted in red in the revised manuscript for clarity.
1. A definition of micropeptides based on size. Some of the discussion involves peptides that are 132 residues long, are those considered micropeptides?
[AU] We appreciate the reviewer’s request for clarification regarding the size-based definition of micropeptides. The 100-amino-acid threshold has historically served as a basic criterion for gene annotation. However, even when this criterion was met, it was often difficult to determine whether the predicted peptides were actually translated and functionally active. As a result, many proteins within this ambiguous size range went unrecognized due to the technical limitations of earlier methods. We have added the following clarification to the revised manuscript to address this point:
Line 101-109: “With these technological advancements, open reading frames (ORFs) ranging from 100 to 300 amino acids previously classified as noncoding due to lack of translational evidence have been found to be actively translated, producing functional proteins that are now recognized as micropeptides. This challenges the traditional size-based definition and underscores the need to redefine micropeptides not solely by their length, but as small, functional proteins that were previously difficult to detect using conventional methods”
2. Some background on the history of gene and protein characterization would be warranted early on in the paper. It is my understanding that at the time of gene cloning (at its peak in the 80s and 90s), any ORFs predicted to generate short peptides (<100 residues maybe?) were disregarded as not functional, despite the fact that short peptides, such as the atrial natriuretic peptide, were already known to have functions and were considered outliers at the time. The same goes for Xist and lncRNAs. This sort of perspective would make the paper more informative in terms of understanding what the limitations and barriers have been in the field of identifying and characterizing micropeptides in the past.
[AU] Thank you for the reviewer’s suggestion. In our review, we have noted that during the early stages of genome annotation, ORFs shorter than 100 amino acids were largely excluded due to the high likelihood that such short sequences arise randomly and do not encode functional proteins. Additionally, while atrial natriuretic peptide is indeed very small in size (28 amino acids), it is generated through post-translational processing of a larger precursor protein that exceeds 100 amino acids in length. Therefore, we believe it should not be considered as an example of a protein derived from an sORF. Similary, Xist is a lncRNA that has not been reported to encode a functional micropeptide, and thus we consider it unrelated to the scope of this manuscript.
3. The authors do not discuss the potential of circRNAs to generate peptides, which has been a proposed functional mechanism for this group of lncRNAs.
[AU] We appreciate the reviewer's insightful comments regarding circRNA-derived micropeptides. While there is emerging evidence supporting their translation and functional roles, this manuscript specifically focuses on micropeptides derived from lncRNAs, rather than circRNAs. Given the distinct differences in biogenesis and structural features that differentiate circRNAs from lncRNAs, we have chosen to limit our discussion to lncRNA-derived micropeptide to maintain a clear and coherent scope. To clarify this point, we have added the following statement in the revised masnucript:
Line 110-117: “Several terms have been used in the literature to describe sORF-derived proteins, including small proteins, microproteins, small or short ORF-encoded peptides/proteins (SEPs) and micropeptides. These micropeptides may originate from lncRNAs, circular RNAs or unannotated sORFs within protein-coding genes, untranslated regions, or intergenic regions 27-30. However, this review focuses soly on micropeptides derived from lncRNAs due to their distinct biogenesis and regulatory features, allowing for a clear and focused discussion.”
4. There are a couple of published papers that were likely not yet out when the authors were preparing their manuscript, but they describe recent advances in nanopore-based sequencing of peptides. One was published in Chemical Science by a French group, and one was a review by a Dutch group published in Nature Biotechnology. Discussion of these papers in the “Perspectives” section would make the manuscript more current. Full disclosure: I am not affiliated with either of these groups.
[AU] We appreciate the reviewer's suggestion to include recent advances in nanopore-based peptide sequencing in the revised manuscript. As suggested, we have added the following text:
Line 591-595: “recent developments in nanopore-based peptide sequencing are revolutionizing the identification of micropeptides. Iesu et al. demonstrated single-molecule resolution capable of distinguishing cis/trans proline isomers, underscoring the potential of nanopores for high-precision peptide analysis”
Reviewer 2 Report
Comments and Suggestions for Authors
The manuscript "LncRNA-Encoded Micropeptides: Expression Validation, Translational Mechanisms, and Roles in Cellular Metabolism" by Ho et al. deals with long non-coding RNA (LncRNA)-derived small proteins. Traditionally, LncRNA has been understood as non-protein coding. However, after almost two decades of intense research, it is now well established that some LncRNAs in fact encode for (small) proteins.
The authors aimed to assemble a comprehensive review covering methods for the detection, validation of stable expression and functional characterization of such micropeptides. Moreover, the review contains a discussion on the issue of translation initiation for non-capped LncRNA and a compendious summary of the physiological roles of those small proteins.
This work provides a valuable overview on LncRNA-derived micropeptides and their functions in eukaryotic (mostly human) cells. However, prior to publication, a few aspects should be addressed:
- The methodological aspects of detection and functional characterization (section 2) are covered much more shallowly than the functional aspects. For instance, several manuscripts published during the last approximately 8 years describing the required adjustments to the default methodologies in RiboSeq and proteomics to cover small proteins. Since that section also contains some questionable information (e.g. labeling techniques such as SILAC do not directly increase sensitivity of micropeptide detection a priori but only provide more accurate quantitative information), and in order to streamline the manuscript, I would suggest only mentioning these aspects briefly in the introduction or perspectives sections. Instead, a more intense discussion on the methods available for functional characterization, including some examples of application, and the specific considerations required with respect to small proteins, would be beneficial.
- Line 72: Micropeptides is one name commonly used for these proteins, but several other names are in use as well (small proteins, microproteins, sORF-encoded-proteins/peptides, SEPs, …). While those proteins may not derive from LncRNA, their similar properties render several considerations applicable to them as well. It would thus be helpful to add a short paragraph mentioning these different names.
- Line 84: The references provides here, do not completely fit. Two of them refer to coli examples and do address LncRNAs but solely any small proteins. Specifically, the differences between eukariotic and prokariotic translation initiation should be made more clear throughout the manuscript and it should become more clear when which organism is mentioned.
- Line 412: There is a space missing between of and cellular.
Author Response
Reviewer 2:
The manuscript "LncRNA-Encoded Micropeptides: Expression Validation, Translational Mechanisms, and Roles in Cellular Metabolism" by Ho et al. deals with long non-coding RNA (LncRNA)-derived small proteins. Traditionally, LncRNA has been understood as non-protein coding. However, after almost two decades of intense research, it is now well established that some LncRNAs in fact encode for (small) proteins.
The authors aimed to assemble a comprehensive review covering methods for the detection, validation of stable expression and functional characterization of such micropeptides. Moreover, the review contains a discussion on the issue of translation initiation for non-capped LncRNA and a compendious summary of the physiological roles of those small proteins.
This work provides a valuable overview on LncRNA-derived micropeptides and their functions in eukaryotic (mostly human) cells. However, prior to publication, a few aspects should be addressed:
[AU] We appreciate the reviewer’s evaluation. We have addressed each comment accordingly, as detailed below. The corresponding revisions have also been highlighted in red in the revised manuscript for clarity.
1. The methodological aspects of detection and functional characterization (section 2) are covered much more shallowly than the functional aspects. For instance, several manuscripts published during the last approximately 8 years describing the required adjustments to the default methodologies in RiboSeq and proteomics to cover small proteins. Since that section also contains some questionable information (e.g. labeling techniques such as SILAC do not directly increase sensitivity of micropeptide detection a priori but only provide more accurate quantitative information), and in order to streamline the manuscript, I would suggest only mentioning these aspects briefly in the introduction or perspectives sections. Instead, a more intense discussion on the methods available for functional characterization, including some examples of application, and the specific considerations required with respect to small proteins, would be beneficial.
[AU] We appreciate the reviewer’s feedback. In response to the comments, we have incorporated the methodological aspects of micropeptide detection into the introduction of the revised manuscript, as below.
Line 92-101: “Advances in ribosome profiling, proteomics, and mass spectrometry particularly the optimized application of improved ribosome footprint generation protocols, and refined mass spectrometry workflows have significantly enhanced the detection and annotation of small proteins. Stable Isotope Labeling with Amino Acids in Cell Culture (SILAC) has improved quantitative accuracy, while complementary approaches such as Data-Independent Acquisition (DIA) mass spectrometry and isobaric tagging methods (iTRAQ, TMT) have expanded detection capabilities and refined the classification of micropeptides.”
We have also added some examples of application of functional analysis of lncRNA-derived micropeptides as below.
Line 274-281: “For instance, HOXB-AS3, a micropeptide encoded by the HOXB-AS3 lncRNA, has been validated through functional studies utilizing CRISPR-Cas9 knockout and overexpression assays, which have confirmed its role in the regulation of tumor growth. Another notable example is SPAR (Small Regulatory Peptide of Amino Acid Response), derived from LINC00961. RNAi knockdown experiments have demonstrated that the depletion of SPAR results in disrupted muscle homeostasis, highlighting its critical regulatory function.”
Line 314-318: “For example, Chu et al. utilized the APEX2 proximity labeling technique to demon-strate that the 123-amino acid micropeptide encoded by C11orf98 sORF is in close proximity to nu-cleophosmin (NPM1) and nucleolin (NCL)”
2. Line 72: Micropeptides is one name commonly used for these proteins, but several other names are in use as well (small proteins, microproteins, sORF-encoded-proteins/peptides, SEPs, …). While those proteins may not derive from LncRNA, their similar properties render several considerations applicable to them as well. It would thus be helpful to add a short paragraph mentioning these different names.
[AU] We appreciate the reviewer’s suggestion. As suggested, we have added the following text in the revised manuscript.
Line 110-117: “Several terms have been used in the literature to describe sORF-drived proteins, including small proteins, microproteins, small or short ORF-encoded peptides/proteins (SEPs) and micropeptides. Some micropeptides originate from lncRNAs or circular RNAs, while others derive from unannotated sORFs within protein-coding genes, untranslated regions, or intergenic regions. However, this review focuses soly on micropeptides derived from lncRNAs due to their distinct biogenesis and regulatory features, allowing for a clear and focused discussion.”
3. Line 84: The references provides here, do not completely fit. Two of them refer to coli examples and do address LncRNAs but solely any small proteins. Specifically, the differences between eukariotic and prokariotic translation initiation should be made more clear throughout the manuscript and it should become more clear when which organism is mentioned.
[AU] We appreciate the reviewer’s feedback. In response to the comments, we have revised the manuscript to clarify that our primary focus is on lncRNA-derived micropeptides in eukaryotic systems. Accordingly, we have removed unrelated references and emphasized the distinctions in eukaryotic translation mechanism.
Line 118-129: “Translation of micropeptides from sORFs in lncRNAs generally follows the canonical mechanisms of eukaryotic mRNA translation, including cap-dependent initiation and ribosome scanning. However, the specific regulatory mechanisms controlling sORF translation in lncRNAs involve non-canonical translation initiation site or interactions with specific RNA-binding proteins and noncoding RNAs. Emerging evidence suggests that some lncRNA-derived micropeptides may be selectively translated under cellular stress condition. However, the underlying mechanisms that govern their translation and biological functions in response to cellular stress remain poorly understood. Given the complexity of translational control, further studies are needed to elucidate the regulatory mechanisms of lncRNA translation.”
4. Line 412: There is a space missing between of and cellular.
[AU] Thank you for bringing this error to our attention. We have corrected it in the revised text.